The effect of uncorrected ametropia on ocular torsion induced by changes in fixation

Oh Kwang-Keun
http://orcid.org/0000-0003-0645-4938 Moon Byeong-Yeon
http://orcid.org/0000-0002-8267-3801 Cho Hyun Gug
http://orcid.org/0000-0001-6806-3305 Kim Sang-Yeob
Yu Dong-Sik yds@kangwon.ac.kr
Department of Optometry, Kangwon National University , Samcheok, Gangwondo , Republic of Korea
Lai Jui-Yang
Electronic publication date: 2021 Aug 4
Publication date: 2021
Volume: 9
Electronic Location ID: e11932
Received 2021 Mar 26; Accepted 2021 Jul 18
Copyright: © 2021 Oh et al.
Copyright year: 2021
Copyright holder: Oh et al.
License: This is an open access article distributed under the terms of the Creative Commons Attribution License, which permits unrestricted use, distribution, reproduction and adaptation in any medium and for any purpose provided that it is properly attributed. For attribution, the original author(s), title, publication source (PeerJ) and either DOI or URL of the article must be cited.
License URL: https://creativecommons.org/licenses/by/4.0/

Keywords: Ocular torsion, Iris images, Uncorrected refractive error, Fixation distances, Slitlamp biomicroscope

Funding: The authors received no funding for this work.

==============================
Background and Objective

Ocular torsion, the eye movements to rotating around the line of sight, has not been well investigated regarding the influence of refractive errors. The purpose of this study was to investigate the effect of uncorrected ametropia on ocular torsion induced by fixation distances.

Methods

Seventy-two subjects were classified according to the type of their refractive error, and ocular torsion of the uncorrected eye was compared based on changes induced by different fixation distances. Ocular torsion was measured using a slit-lamp biomicroscope equipped with an ophthalmic camera and a half-silvered mirror.

Results

In all groups, excyclotorsion values increased as the fixation distance decreased, but the myopia and astigmatism groups had larger amounts of ocular torsion than the emmetropia group. In addition, as the amount of uncorrected myopia and astigmatism increased, the amount of ocular torsion increased.

Conclusion

Since the amount of ocular torsion caused by a change to a shorter fixation distance was larger when the refractive error was uncorrected, we suggest that ametropia should be fully corrected in patients frequently exposed to ocular torsion due to changes in fixation distance.

Introduction

For clear visual perception, the retinal correspondence of both eyes prevents diplopia and enables stereoscopic vision. When moving the fixating object, the image of the retina moves quickly causing disturbances to the image. Therefore, eye movement is required for stable binocular retinal correspondence (Van Rijn, Van der Steen & Collewijn, 1994). Eye movement is divided into three components: horizontal, vertical, and torsional movements (Van Rijn, 1994). Ocular torsion, also known as rotation, is divided into excyclotorsion and incyclotorsion for outward and inward rotation, respectively, above the vertical meridian of the cornea (Sheiman & Wick, 2014).

Ocular torsion may occur as a compensatory movement that stabilizes the image of the retina in an environment such as in response to a head tilt (Schworm et al., 2002; Ko et al., 2011). This ocular counter-roll is generated by the vestibular ocular reflex (Schworm et al., 2002) and may be caused by signal imbalances of the vertical semicircular canals, but it is mainly induced by the action of the otolith organs (De Graaf, Bos & Groen, 1996).

The ocular torsion can also occur with a static posture based on the distance of the fixation target (Allen, 1954; Enright, 1980). Ocular torsion in static posture occurring without tilting of the head or body is one of the factors that change the astigmatic axis between distant and near fixation (Leonard Werner & Press, 2002). Scobee (1952) reported that the astigmatic axis changed in 77% of 247 examined subjects when the fixation was moved from a distant to a near point, and reported that it changed up to 10°. Although the rate of change in the astigmatic axis is large, the prescription rate for symptom relief is actually small. In the study by Joo & Sim (2007), 837 (52.3%) out of 1598 eyes showed changes of more than 10° in the astigmatic axis. There are several factors that temporary change the distant and near axis of astigmatism. These factors include accommodation (Joo & Sim, 2007; Hughes, 1941; O’brien & Bannon, 1948), size of the pupil (Chikako, Yoshiko & Ikuko, 2004; Asharlous et al., 2016; Noh et al., 2014), and the excyclotorsion effect of near vision (Allen, 1954; Leonard Werner & Press, 2002; O’brien & Bannon, 1948). Changes in the axis of astigmatism occurring at distant and near distance can affect the visual acuity. Visual acuity, stable fixation and eye movement have a great influence on the visual perception (Martinez-Conde, Macknik & Hubel, 2004). There are several previous studies on visual perception and eye movements (Westheimer & McKee, 1975; Gegenfurtner, 2016; Ibbotson & Krekelberg, 2011); however, relatively few studies have examined among ocular torsion. The study by Van Rijn, Van der Steen & Collewijn (1994) suggests that the instability of ocular torsion mainly affects the periphery of the retina. Therefore, they reported that studies of torsional movements are relatively few compared to horizontal and vertical movements. Baskaran et al. (2019) reported less torsional vergence amplitude in myopes than in emmetropes. Moreover, in several previous studies, the uncorrected refractive error resulted in unstable fixation (Ko, Snodderly & Poletti, 2016; Yu et al., 2017; Wahl, Dragneva & Rifai, 2019); when the stimulus of fixation was weak, the amount of ocular torsion increase (Choi et al., 2006). These previous studies indicate that uncorrected ametropia may affect ocular torsion.

Since ocular torsion induced by changes in fixation distance will influence the astigmatic axis and affect visual acuity and subjective symptoms, quantitative ocular torsion angle measurement is important. There are several methods for measuring ocular torsion. An indirect ophthalmoscope (Madigan & Katz, 1992), video and digital camera (Bos & De Graaf, 1994; Ong & Haslwanter, 2010; Felius et al., 2009), slit-lamp biomicroscope (Jethani et al., 2010), and fundus photography are used to measure ocular torsion. However, there is no standardized approach. Among the techniques for measuring ocular torsion, the most commonly used method is fundus photography (Jethani et al., 2010; Williams & Wilkinson, 1992; Rosenbaum & Santiago, 1999; Kothari et al., 2005). However, since the distance of the target is fixed in fundus photography, as well as in the above-mentioned methods, it is difficult to assess ocular torsion induced by various fixation distances. In the present, fundus photography has a fixation target fixed inside and outside, or there is no fixation target (Panwar et al., 2015). Therefore, the amount of change in ocular torsion for a specific range of fixation distance cannot be measured. To measure this effect, we thought that specular reflection may be useful. If the fixation distance is shorter than 5–6 m, specular reflection is a method that is often used for subjective or objective refraction in practice (Andrew & Caroline, 2007). In addition, in the case of simple myopia or hyperopia without astigmatism, the measurement of ocular torsion induced by a change in the fixation distance is difficult since there is no reference marker for rotation, unlike in astigmatism. However, we expected that ocular torsion in eyes without astigmatism can be measured using crypts on iris images. Using slit-lamp biomicroscopy and a half-silvered mirror, it may be possible to measure ocular torsion at any fixation distance.

The purpose of this study was to investigate the influence of uncorrected ametropia on ocular torsion induced by changes in fixation distance using crypts on iris images taken by slit-lamp biomicroscopy with a half-silvered mirror.

Materials and Methods

Paricipants

A total of 72 college students (42 men, 30 women, mean age: 23.07 ± 1.87 years) without ophthalmic and vestibular diseases participated in this study. Among the subjects, there were 12 subjects with emmetropia, 57 subjects with myopia, and three subjects with hyperopia. None of the subjects had a history of strabismus or ocular surgery, as well as a history of disorders affecting the cornea or iris. The amplitude of accommodation using push-up method was conducted to exclude the instability of fixation at near owing to an accommodative dysfunction. The mean accommodation of subjects participating in this study was 10.84 ± 2.35 D, and there were no subjects with accommodative dysfunction such as insufficiency or excess of accommodation and insufficiency of accommodation facility. In addition, it was confirmed that there was no dysfunction in the extraocular muscle and eye movement by performing the extraocular motility test on all subjects. The subjects with amblyopia were excluded from this study. This study was approved by the Institutional Review Board of Kangwon National University (IRB approval number: KWNUURB-2020-06-007-001) and adhered to the tenets of the Declaration of Helsinki. All participants provided written informed consent.

Experimental protocol

The procedure of this study is shown in Fig. 1. After determining the refractive error of the study participants by subjective refraction using a manual phoropter (Phoropter 11625B; Reichert, Depew, NY, USA), the iris images of the subjects were taken with a slit-lamp biomicroscope (SL-D701; Topcon, Japan) equipped with an ophthalmic camera (DC-4; Topcon, Japan) and a half-silvered mirror (Fig. 2). The iris images were photographed three times at 16×g magnification at a fixation distance of 5 m, 3 m and 40 cm, and images with the most distinctive features, such as iris crypts, were analyzed using the ImageJ program (version 1.52a, National Institutes of Health, Bethesda, MD, USA) (NIH, 2011). The fixation target was point target, and an iris image was taken in a binocular fixating condition. The diameter of the point target was 2.5 cm (visual angle of 0.29 degrees at 5 m and 0.35 degrees at 40 cm). In order to exclude the effect of convergence that occurs as the near fixation distance, the specific image (Fig. 3) was attached to the computer monitor, and the camera was moved slightly inward to keep the front of the pupil as much as possible. In the iris images, the iris crypts of the upper pupil were used as a reference point, and the iris crypts of the lower portion of the pupil were connected through the pupil center as much as possible, and a horizontal line was drawn to measure the angle. Ocular torsion was calculated as the difference between the angle at the 5-m fixation distance (reference point) and the angle at the other fixation distances (3 m or 40 cm), as shown in Fig. 4. The ocular torsion measurement method used in this study was introduced in the previous study and was reported to be suitable for measuring the ocular torsion (Oh et al., 2021).

Figure 1 Flow chart for the ocular torsion of uncorrected refractive error.

Figure 2 Slitlamp biomicroscope using a camera and a half-silvered mirror.

(A) View from the side. (B) View from above.

Figure 3 The specific image using on this study.

(A) The specific image. (B) The specific image attached to the monitor.

Figure 4 Ocular torsion of a left eye determined with ImageJ.

The fixation distances are 5 m (left image) and 40 cm (right image). In this case, ocular torsion is 1.731° (97.157°–95.426°) excyclotorsion.

The determined ocular torsion angle was analyzed according to the type and amount of ametropia. Referring to previous studies, emmetropia was within ±0.50 D (Shim, Shim & Joo, 2006), whereas myopia was divided into a low group (−3.00 D or below), a middle group (−3.25 D to −6.00 D), and a high group (over −6.25 D) (Chebil et al., 2015; Kapadia & Wilson, 2000). Astigmatism was analyzed by dividing the study population into a −1.00 D or less group and a −1.25 D or more group (Villegas, Alcón & Artal, 2014).

Data analysis

All the data collected were analyzed using IBM SPSS statistics software (version 24.0; IBM Corp., Armonk, NY, USA). To confirm the change of ocular torsion induced by fixation distance, the paired samples t-test with Bonferroni post-hoc, which is a parametric test, was performed. In addition, Mann–Whitney U test, which is a non-parametric test, was conducted for comparisons between the emmetropia and myopia groups. Kruskal–Wallis test for non-parametric analysis of variance was used for the comparative analysis between the ocular torsion angle and groups of ametropia. Pearson’s correlation test determined the correlation between the amount of ametropia and ocular torsion. A p-value of ≤0.05 was considered significant in this study.

Results

Participants characteristics

Their mean refractive power was −3.17 ± 2.97 D for spherical equivalent and −0.86 ± 0.75 D for cylindrical errors. A detailed analysis of the participant’s refractive errors revealed that, the range of refractive error was −0.16 ± 0.21 D in 12 participants (mean age: 22.67 ± 1.72 years) with emmetropia, and the range of refractive error was –1.79±1.26 D in eight participants (mean age: 23.13 ± 1.73 years) with simple myopia. There were 24 participants (mean age: 22.88 ± 1.80 years) with an astigmatism amount of 1.00 D or less, and the range of refractive error was −4.36 ± 2.55 D. The number of participants with an amount of astigmatism 1.25 D or greater was 25 (mean age: 23.16 ± 1.62 years), and, the range of refractive error was −5.02 ± 2.69 D. The mean lateral phoria measured by the Maddox rod test was 1.78 ± 4.20 and 5.67 ± 7.19 prism diopter (∆) exophoria at distant and near, respectively, and the mean vertical phoria was 0.22 ± 1.17 ∆ hypophoria (for the right eye as a reference) at both distance and near. Cyclophoria was excluded in all study subjects using the Double Maddox rod test.

Comparison of ocular torsion at fixation distances

The mean ocular torsion values measured at fixation distances of 40 cm and 3 m are shown in Table 1. In both eyes, as the fixation distance decreased, the excyclotorsion increased (p < 0.001 for both eyes, paired t-test). The mean difference in excyclotorsion between the right eye and the left eye was 0.05° at 40 cm and 0.07° at 3 m, and the left eye had a higher rotation tendency than the right eye, but this difference was not statistically significant (p = 0.738 at 40 cm, p = 0.500 at 3 m, paired t-test). In the present study, a dominance of the right eye was observed in 53 subjects (73.6%), and a left dominant eye was present in 19 subjects (26.4%).

Table 1 Mean ocular torsion induced by different fixation distances.

Fixation distance	Right eye	Left eye	
40 cm	1.76 ± 0.90°	1.81 ± 1.21°	
3 m	0.97 ± 0.68°	1.04 ± 0.88°	
Mean difference	0.79°	0.77°	
Paired t-test (p)	<0.001	<0.001	
n	72	72	

Figure 5 shows the mean excyclotorsion of the dominant and non-dominant eye induced by different fixation distances. The dominant eye had lower mean excyclotorsion values at both 40 cm and 3 m than the non-dominant eye but did not show a significant difference (p = 0.470 at 40 cm, p = 0.577 at 3 m, paired t-test). Therefore, the analysis according to the type of ametropia was based on the dominant eye.

Figure 5 Comparison of the ocular torsion between the dominant and non-dominant eye at different fixation distances.

The number of subjects with the right dominant eye was 53, and the left dominant eye was 19.

Difference of ocular torsion between myopia and emmetropia

In all groups of defined ametropia types, the mean ocular torsion increased with decreasing fixation distance. In the hyperopic group, the ranges of mean excyclotorsion at 40-cm and 3-m fixation distances were 1.65–2.46° and 1.12–1.28°, respectively, but the number of subjects was too small (three subjects), so they were excluded from further analyses in this study. Therefore, Table 2 shows only the changes in ocular torsion at different fixation distances in the emmetropic and myopic groups, excluding the hyperopic group. There was no significant difference in the mean ocular torsion between the emmetropia and myopia groups at both 40-cm and 3-m fixation distances (p = 0.062 at 40 cm, p = 0.178 at 3 m, paired t-test), but the myopia group showed a slightly larger excyclotorsion than the emmetropia group.

Table 2 Changes in ocular torsion at different fixation distances according to uncorrected myopia.

Fixation distance	Emmetropia (n = 12)	Myopia (n = 57)	p†	
40 cm	1.26 ± 0.78°	1.80 ± 0.92°	0.062	
3 m	0.74 ± 0.70°	1.00 ± 0.69°	0.178	
Mean difference	0.52°	0.80°	–	
Notes:

† p-value for Mann–Whitney U test.

Ocular torsion values were determined based on the dominant eye. The refractive state was analyzed with the spherical equivalent.

Ocular torsion according to the amount of myopia

Figure 6 shows the analysis of the mean ocular torsion according to the amount of myopia at different fixation distances. The mean excyclotorsion values at the fixation distances of 40 cm and 3 m in the myopia of −3.00 D or less group (n = 22) were 1.31 ± 0.63° and 0.72 ± 0.52°, respectively. In the myopia group of −3.01 to −6.00 D (n = 20), the mean excyclotorsion values were 1.82 ± 0.94° and 1.03 ± 0.56°, respectively, and 2.56 ± 0.81° and 1.42 ± 0.89°, respectively, in the myopia of −6.01 D or higher group (n = 15). The mean excyclotorsion increased as the amount of myopia increased at both 40-cm and 3-m fixation distances (p < 0.001 at 40 cm, p = 0.015 at 3 m, Kruskal–Wallis test). In addition, there was a significant correlation between the amount of myopia and the ocular torsion angle at a 40-cm fixation distance (r = −0.320, p = 0.006, y = 1.18 − 0.15 ×), but no significant correlation at the 3-m fixation distance (r = −0.225, p = 0.057, y = 0.69 − 0.08 ×).

Figure 6 Mean ocular torsion according to the amount of myopia at different fixation distances.

Upper and lower lines indicate linear regression at 40 cm and 3 m fixation distance, respectively.

Ocular torsion according to myopic astigmatism

Figure 7 shows the changes in ocular torsion according to the amount of astigmatism at 40-cm and 3-m fixation distances. The mean excyclotorsion in the non-astigmatism group (n = 20) was 1.41 ± 0.74° and 0.81 ± 0.53° at 40-cm and 3-m fixation distances, respectively, 1.52 ± 0.86° and 0.87 ± 0.69° in the −1.00 D or less group (n = 24), and 2.19 ± 0.92° and 1.21 ± 0.73° in the −1.25 D or higher group (n = 25). As the amount of astigmatism increased, the mean excyclotorsion also increased; however, only the 40-cm fixation distance showed a significant difference (p = 0.008, Kruskal–Wallis test). As a result of the correlation analysis between the amount of astigmatism and ocular torsion, there was a significant correlation in both 40-cm and 3-m fixation distances (r = −0.440, p = 0.001, y = 1.41 − 0.39 × at 40 cm; r = −0.288, p = 0.030, y = 0.81 − 0.21 × at 3 m).

Figure 7 Mean ocular torsion according to the amount of astigmatism at different fixation distances.

Upper and lower lines indicate linear regression at 40 cm and 3 m fixation distance, respectively.

Discussion

In this study, we measured ocular torsion induced by varying fixation distances and analyzed it according to the type of ametropia. Our main finding was that in shorter fixation distance, no statistically significant difference between the emmetropia group and the myopia group was observed; however, the angle of ocular torsion was larger as the amount of myopia or astigmatism increased.

Eye movements comprise horizontal, vertical, and torsional movements. Ocular torsion is caused by head tilt, among other stimuli. We anticipated that ocular torsion without a head tilt would be the result of a change in fixation distance, and our findings showed that the shorter the fixation distance, the more excyclotorsion occurred. Ocular torsion occurs depending on the fixation distance (Allen, 1954; Leonard Werner & Press, 2002). In a previous study on ocular torsion induced by varying fixation distances (Allen, 1954), it was reported that ocular torsion does not occur when performing horizontal conjugate movements, but excyclotorsion can be observed during horizontal disjunctive (i.e., convergence) or elevation movements. This action is related to the activation of the third cranial nerve nucleus stimulating the medial rectus muscle and the extraocular muscles (superior rectus, inferior rectus, inferior oblique) innervated by this cranial nerve (Allen, 1954). Therefore, it has been reported that the main action of the superior and inferior rectus muscles is neutralized, and excyclotorsion occurs due to the influence of the inferior oblique muscles (Mays et al., 1991). Due to this action, Allen & Carter (1967) recommended that excyclotorsion should be added to the near reflex (accommodation, convergence, myosis) that occurs during near fixation.

Our results showed that ocular torsions were excyclotorsions of 1.76–1.81° for 40 cm and 0.97–1.04° for 3 m. The mean pupillary distance of the subjects participating in this study was 61.93 mm, and when the convergence angle for the 40-cm fixation was calculated from this, a value of approximately 7.74° (half of 15.48°) was obtained. Landolt (1876) reported that an excyclotorsion angle of 3.4° was shown at a convergence angle of 15°. Referring to this, at a convergence angle of 7.5°, an excyclotorsion angle of 1.7° can be estimated. In the study by Allen & Carter (1967) using a reflex camera for subjects with unknown refractive errors, an ocular torsion angle of 4.30° was observed for a convergence angle of 18° at 0° elevation angle, and it was reported that an ocular torsion angle of 0.24° occurred at a 1° convergence angle. Applying our results to this study of Allen and Carter, the estimated excyclotorsion was approximately 0.23° at a 40-cm fixation distance, very similar to that reported by Allen and Carter. Another study on excyclotorsion associated with convergence in monkeys reported an ocular torsion angle of 1.05° at an 18.2° convergence angle, which is very different from the present study (Mays et al., 1991). Such differences in subjects and measurement methods could account for the similarities and differences seen in our results compared to previous studies.

In the current study, the amount of ocular torsion also increased as the amount of myopia increased. Ocular torsion that occurs in eyes with uncorrected myopia will change the astigmatic axis, and it is thought that the larger the amount of uncorrected myopia, the greater the amount of astigmatic axis changes. In previous studies on myopia and fixation stability (Zhu et al., 2019), it was reported that myopia led to eye movements due to unstable retinal images, and increased eye movements were observed in myopia compared to emmetropia. In addition, Choi et al. (2006) reported that ocular torsion occurred less often in a stable fixation or stronger fixation state. Based on these previous studies, it is thought that more ocular torsion will occurs due to unstable retinal image, and it is expected that the amount of ocular torsion in the myopia group is larger than that in the emmetropia group. It has been reported that patients with high myopia have poor fixation and generate eye movements such as microsaccades, tremor, and drift during fixation (Ko, Snodderly & Poletti, 2016; Yu et al., 2017). Therefore, it is expected that patients with high myopia or astigmatism will experience discomfort and subjective symptoms due to changes in the astigmatic axis when under- or uncorrected.

In our results, uncorrected astigmatism increased the amount of ocular torsion induced by changes in fixation distance. Ozulken & Ilhan (2019), who studied the relationship between astigmatism and ocular torsion before and after photorefractive keratectomy, reported that high astigmatism before surgery was related to large ocular torsion, and this also affected postoperative measurements. In another study, it was suggested that changing the power and axis of astigmatism decreased the quality of the retinal image (Pujol et al., 1998). Therefore, under- or uncorrected astigmatism power and axis will reduce the quality of the retinal image, and it is thought that the larger the amount of uncorrected astigmatism, the greater the amount of ocular torsion required to stabilize the inferior retinal image.

In the results of this study, the shorter the fixation distance, the more excyclotorsion occurred, and the amount of excyclotorsion increased more when myopia and astigmatism were not corrected. Changes in the astigmatic axis caused by ocular torsion will result in decreased visual acuity, discomfort, fatigue, and various subjective symptoms. These will be more severe as the amount of uncorrected astigmatism increases. In addition, it is expected that an increase in unnecessary ocular torsion will potentially cause a small amount of cyclophoria. Wick & Ryan (1981) reported that the range of normal cyclophoria at a distance of 6 m was approximately 0.75 ± 1.15°, which differs from the ocular torsion angle caused by uncorrected ametropia at a fixation distance of 40 cm in the present study. Howe (1907) implied that the small amount of excyclophoria at near fixation distance were observed in just 25% of the study subjects and probably would not have a clinically affect. However, if the amount of uncorrected ametropia is large, the amount of ocular torsion angle also increases; so it is thought that full correction should be achieved for ametropia. In uncorrected astigmatism, the change in the astigmatic axis due to ocular torsion is expected to cause symptoms similar to cyclophoria in severe cases. Cyclophoria induces subjective symptoms like headaches, burning eye sensations, restless fatigue, slow reading speed, and missing letters within a line when reading (Sheiman & Wick, 2014). In a study evaluating the change in the astigmatic axis and quality of the retinal image (Pujol et al., 1998), it was reported that the in vivo eye, unlike the theoretical model eye, is somewhat corrected by aberration when the change in the astigmatic axis is small. However, there may be a limit for aberration-induced corrections in uncorrected cases with high amounts of myopia or astigmatism.

This study has several limitations. Among the recruited subjects, the number of subjects with hyperopia was too small. The standard of emmetropia applied in this study was ±0.50 D, and only three subjects had +0.75 D or higher. Therefore, the analysis of the hyperopia group was limited. Moreover, because there were only three subjects with an against-the-rule astigmatic axis and six subjects with an oblique astigmatic axis, the analysis related to the astigmatic axis was limited. Thus, further studies on hyperopia and the astigmatic axis should be conducted. Moreover, in the present study, uncorrected ametropia was not compared with fully corrected ametropia of the same participant, but with emmetropia. However, corrected ametropia is similar in visual function to that of emmetropia (Roch-Levecq et al., 2008), indicating that this should not limit the comparison between the ametropia and emmetropia groups. Another limitation is vergence. In this study, in order to minimize the effect of vergence, participants who were not included in the normal criteria were excluded from this study by conducting the vergence test. Nevertheless, vergence may affect fixation, limiting accurate ocular torsion measurement.

Conclusions

Summarizing the results of this study, when the distance changes from distant (5 m) to near (40 cm or 3 m) fixation, excyclotorsion occurred. The amount of excyclotorsion between the emmetropia group and the myopia group was similar, but as a result of dividing the myopia group into three group, the amount of excyclotorsion increased as the amount of myopia increased. Similarly, excyclotorsion values were increased with increases in the uncorrected amount of astigmatism. This means that in cases with large amounts of uncorrected myopia or astigmatism, ocular torsion will occur during near-distance activities such as reading more pronounced than in fully corrected cases. Therefore, we suggest that a full correction should be performed to reduce changes in ocular torsion due to ametropia, especially when working at near distance.

Supplemental Information

Supplemental Information 1 Raw data.

Click here for additional data file.

Additional Information and Declarations

Competing Interests

Author Contributions

Human Ethics

Data Availability

The authors declare that they have no competing interests.

Kwang-Keun Oh conceived and designed the experiments, performed the experiments, analyzed the data, prepared figures and/or tables, authored or reviewed drafts of the paper, and approved the final draft.

Byeong-Yeon Moon performed the experiments, analyzed the data, prepared figures and/or tables, and approved the final draft.

Hyun Gug Cho analyzed the data, authored or reviewed drafts of the paper, and approved the final draft.

Sang-Yeob Kim analyzed the data, prepared figures and/or tables, and approved the final draft.

Dong-Sik Yu conceived and designed the experiments, performed the experiments, analyzed the data, authored or reviewed drafts of the paper, and approved the final draft.

The following information was supplied relating to ethical approvals (i.e., approving body and any reference numbers):

This study was approved by the Institutional Review Board of Kangwon National University (KWNUURB-2020-06-007-001).

The following information was supplied regarding data availability:

The raw measurements are available in a Supplemental File.

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
