# Peer review of "The effect of uncorrected ametropia on ocular torsion induced by changes in fixation"

_PeerJ, doi:10.7717/peerj.11932_

## Round 0.1 · original submission · Major Revisions

Dear Authors,

Your manuscript has been reviewed by two external experts. The professional reviewers point out some major drawbacks that seriously diminish the research quality and scientific soundness. Before further consideration of your manuscript for publication in PeerJ, all the concerns raised by the referees should be addressed. Thank you very much.

Reviewer 1 ·

Basic reporting

Introduction: There are several points described in the literature, but no further elaboration to inform the details of the cited studies.
1. Line 57: “…, but studies on these changes due to eye movement are insufficient compared to other factors.” This statement is not clear. The authors need to explain what are the factors and the short-coming related to these factors. Please include the references.
2. Line 58: “There are several studies on visual perception and eye movement (Westheimer & McKee, 1975),…” Only one study for this statement. What is the significance of this reference study in relation to your research? This statement needs clarification.
3. Line 59: “…, but relatively few studies have examined ocular torsion.” Please quote the references for this statement.
4. Line 60: “The study by …….of the retina.” The authors relate ocular rotation to the peripheral retina. What is the relation of stimulation of the peripheral retina and the relevance to this study's aim?
5. Line 64: “…, can affect uncorrected ametropia.” Please include references for this statement.
6. Line 67: “Although there are several methods for measuring ocular torsion…”. For this statement, please list the techniques used in the literature with references.
7. Line 71: “However, since the ……..fixation distances”. The statement is not clear. The authors are explaining the weakness of the technique. Perhaps, elaborating the amount of change in ocular rotation for a specific range of fixation distance could better clarify this statement.
8. Line 81: “The method….. iris image (Kothari et al., 2005).” The statement is repeating information in line 67-68. Suggest removing the sentence.

Experimental design

Participants:
1. Please include age information. Although the age factor may not affect the convergence, it affects the accommodation function. A poorer accommodation function has been related to the instability of fixation at near. It could change the exertion from the vergence system, particularly in certain binocular vision anomalies such as the accommodation insufficiency or accommodation excess. Has the accommodation function been considered in participants’ screening?
2. Participants who cannot achieve 6/6 vision with optical correction in each eye and binocularly were eligible for the study? For instance, amblyopic patients and patients with high astigmatism (-2 DC) show poor visual quality and likely affect the visual fixation.

Experiment protocols:
1. Line 109: “The fixation target used point target,…”. When changing fixation to a near object, always near triad happens (miosis, accommodation and convergence) to ensure clear single vision at near. My question, is there any particular reason for using the “point target” as the fixation target? As we know, point target is not a good stimulus for accommodation.
2. The size of the point target is not informed in the article. Did the authors change the point target size when testing different myopia groups?

Data analysis:
1. The analysis section is not clear. Suggest the authors elaborate the purpose of the analysis and inform the analysis tools applied. For example, ”To compare the ocular torsion between emmetropes and myopes, unpaired t-test was conducted….”.

Figure 5
Suggest to include “myopic subjects” and number of subject in the title.

Validity of the findings

Participants characteristics:
1. Please include the number and refractive range of participants with emmetropia, simple myopia, myopia with astigmatism of 1 DC or lesser, and myopia with astigmatism of 1.25 DC or greater.
2. Also, please include the age range for each of the refractive sub-categories

Results: Line 142 – 191
1. Overall, the main objective of the paper is to relate the type of ametropia (emmetropia , simple myopia, myopia with low astigmatism, myopia with high astigmatism) and the ocular rotation at 3m and 40cm. Sub-sections in the results are confusing; without reporting which refractive groups were compared, no information of the number of participants included in the type of ametropia group and the simple t-test limit the testing of the interaction effect of within-subject factors (fixation distance and type of ametropia). Perhaps, performing a repeated measure ANOVA (exclude the 3 hyperopes) could simplify the analysis and better present the results.
2. Based on analysis in point 1, if the type of ametropia did not change the ocular rotation significantly, then, the data can be grouped to test the effect of ocular dominancy and ocular rotation.
3. Line 148: “It has been reported that……” This statement is inappropriate in the result section. Please move to the discussion.
4. Line 151 – 152: This statement does not add additional information and redundant to the statement in line 154. Suggest removing the statement.

Discussion:
1. Line 195 – 197: The results showed the ocular rotation was not statistically significant between emmetropia and myopia. But, the statement reported myopia and astigmatism group showed larger ocular rotation. This statement is inappropriate unless the authors could relate the minimum angle of rotation that impacts the visual functions.
2. Line 254: “….increased more when ametropia and astigmatism…”. The word “ametropia” is not appropriate since the authors compared the emmetropia and myopia group. Suggest changing it to “myopia”.

Conclusions:
Line 292 – 293: “Therefore, we suggest that a full correction should be performed ……near distance.” I am concern with this statement. Myopia of 3 D or greater is rarely left uncorrected for distance and near vision in real-world practice. What is the significance of this study’s results in relation to clinical practice?

·

Basic reporting

no comment

Experimental design

no comment

Validity of the findings

- One potential confound in the article is the amount of vergence across distances. I imagine this is something that it is not possible to measure at this stage but at least it should be addressed in the discussion as a limitation. It could be that some subjects were more accurately fixating binocularly in the target and thus inducing a different amount of torsion.
- Throughout the authors propose the idea that torsion is somehow used to stabilize the retinal image that moves due to eye movements. I cannot follow the logic of the argument as stated right now in the manuscript.

Additional comments

- Abstract and line 202: It is not correct to say "Ocular torsion, one of the eye movements to stabilize the
retinal image". Ocular torsion is a more generic term simply referring to eye movements when the eye rotates around the line of sight.
- Lines 34-35: These two sentences appear to contradict each other. Appears to say something like, "Eye movement is required .... because eye movements move the retina"?
- Line 35: I would not use the word type I would say components or dimensions.
- Line 42: the sentence "but it can also occur with a static posture depending on the distance of the fixation target" seems a bit out of place. Specially since the next sentence uses the term ocular counter-roll. The next paragraph seems a more appropriate place to introduce torsion due to posture.
- Line 61: Do you mean the studies are smaller (fewer) or that the torsional movements are smaller?
- Line 63: How do torsional eye movement stabilize the image following changes in fixation distance?
- Line 74-83. It is very hard to follow which methods you propose and which ones you actually used?
- Line 109: The sentence seems incomplete "used point target".
- Line 157: I don't understand the logic for using only the dominant eye data. Why not averaging both, for example?
- Line 237: Similar to a previous comment. I don't understand the idea of torsion stabilizing the retinal image that moves due to eye movements.

---

## Round 0.2 · accepted · Accept

Dear Authors,

The revised manuscript is now suitable for publication. Once again, thank you very much for considering PeerJ to publish your work.